# Biosynthesis of Metal Nanoparticles via Microbial Enzymes: A Mechanistic Approach

**DOI:** 10.3390/ijms19124100

**Published:** 2018-12-18

**Authors:** Muhammad Ovais, Ali Talha Khalil, Muhammad Ayaz, Irshad Ahmad, Susheel Kumar Nethi, Sudip Mukherjee

**Affiliations:** 1CAS Key Laboratory for Biomedical Effects of Nanomaterials and Nanosafety, CAS Center for Excellence in Nanoscience, National Center for Nanoscience and Technology (NCNST), Beijing 100190, China; 2University of Chinese Academy of Sciences, Beijing 100049, China; 3Department of Eastern Medicine and Surgery, Qarshi University, Lahore 54000, Pakistan; talhakhalil.qau@gmail.com; 4Department of Pharmacy, University of Malakand, Khyber Pakhtunkhwa (KPK), Chakdara 18000, Pakistan; ayazuop@gmail.com; 5Department of Life sciences, King Fahd University of Petroleum and Minerals (KFUPM), Dhahran 31261, Saudi Arabia; irshad@kfupm.edu.sa; 6Department of Experimental and Clinical Pharmacology, College of Pharmacy, University of Minnesota, Minneapolis, MN 55455, USA; susheel.nethi@gmail.com; 7Department of Bioengineering, Rice University, Houston, TX 77030, USA

**Keywords:** metal nanoparticles, microbial flora, biosynthesis, microbial enzymes, action mechanism

## Abstract

During the last decade, metal nanoparticles (MtNPs) have gained immense popularity due to their characteristic physicochemical properties, as well as containing antimicrobial, anti-cancer, catalyzing, optical, electronic and magnetic properties. Primarily, these MtNPs have been synthesized through different physical and chemical methods. However, these conventional methods have various drawbacks, such as high energy consumption, high cost and the involvement of toxic chemical substances. Microbial flora has provided an alternative platform for the biological synthesis of MtNPs in an eco-friendly and cost effective way. In this article we have focused on various microorganisms used for the synthesis of different MtNPs. We also have elaborated on the intracellular and extracellular mechanisms of MtNP synthesis in microorganisms, and have highlighted their advantages along with their challenges. Moreover, due to several advantages over chemically synthesized nanoparticles, the microbial MtNPs, with their exclusive and dynamic characteristics, can be used in different sectors like the agriculture, medicine, cosmetics and biotechnology industries in the near future.

## 1. Background and Role of Microbial Enzymes in Metal Nanoparticle (MtNP) Biosynthesis 

Nanotechnology has accomplished enormous development during the last decades due to the synthesis and varied applications of metal nanoparticles (MtNPs) in different areas, such as biology, food, agriculture, engineering, electronics, cosmetics, medicine, and in food and biomedical devices. MtNPs have reached a momentous position due to their specific physicochemical characteristics and significant biotechnological applications [1,2]. According to a market report, the worldwide production of MtNPs is currently valued at 13.7 billon US dollars, which is expected to reach 50 billon US dollars by 2026. The wide-ranging use of MtNPs has significantly contributed to robust development in the macroeconomic industry and the demand for MtNPs will continue to be high in three regions of the world: North America, Western Europe and the Asia-Pacific region [3].

During the last decade, many researchers have synthesized MtNPs by orthodox physical and chemical methods. The disadvantages of the physical method include expensive synthesis and little yield. Similarly, the chemical methods are unsafe due to the involvement of hazardous chemical substances that are attached to the surface of MtNPs, which possess detrimental side effects in biomedical applications. Keeping in view the aforementioned concerns, research has shifted towards the synthesis of MtNPs using biological constituents which are economical, biocompatible, non-toxic and eco-friendly [4,5,6,7]. The biological synthesis of MtNPs is mostly carried out by utilizing different types of plants and microorganisms. The phytogenic process of MtNP synthesis is economical and relatively simple. However, this process generates polydispersed nanoparticles, due to its diverse photochemistry [8,9]. In contrast, the microorganisms are considered as prospective bio-factories for the green synthesis of MtNPs, something that has gained vast consideration in recent times, as they are indispensable and of technological importance. This is due to the broad range of microorganisms that react in a different way with the metal ions for MtNP synthesis. Microbial MtNPs of different sizes and shapes have been reported in various species of bacteria, fungi and yeasts due to their enhanced growth rate, easy cultivation and their capability to grow under established conditions of temperature, pH and pressure [10,11]. The biosynthesis of MtNPs and their alloys (gold, silver, gold-silver alloy, selenium, tellurium, platinum, palladium, silica, titania, zirconia, quantum dots, magnetite and uraninite) has been reported in different microorganisms, including bacteria, actinomycetes, fungi, yeasts and viruses. 

Different microorganisms have the ability to synthesize inorganic materials via both intracellular and extracellular methods. In the intracellular approach, the microbial cell is comprised of an amazing ion transport system. Due to electrostatic interaction, the bacterial cell wall which is negatively charged attracts the positively charged metal ions. Additionally, the bacterial cell wall contains enzymes that reduce the metal ions to their respective nanoparticles. Whereas in extracellular method, the microbial cell secretes reductases used in the bioreduction of metal ions into the corresponding MtNPs [12]. The following sections of this article will provide an overview on the green synthesis of MtNPs by using potential microbial flora, including bacteria, cyanobacteria, microalgae, actinomycetes, yeasts, fungi, viruses and diatoms. Furthermore, the intracellular and extracellular mechanisms that are adopted by the microorganism and the microbial enzymes for MtNP synthesis will be elaborated upon, along with the contemporary challenges and future prospects. Table 1 highlights the recent literature about the microbe assisted synthesis of MtNPs and their applications.

## 2. Biosynthesis of MtNPs by Microorganisms

Synthesizing uniform, ultrafine, well-dispersed functional nanoparticles under normal conditions through a controlled manner remains a great challenge [13,14]. This has caused a major surge in looking for alternative means of synthesizing nanoparticles that are devoid of such disadvantages. In recent years, biological resources have been frequently explored for the biosynthesis of metal or metal-based nanoparticles. These biological resources usually provide a versatile, economical and eco-friendly method to fabricate metal nanoparticles [14,15] that exhibit interesting physical, chemical and biological properties. Furthermore, other advantages such as the ease of production and scaling, well defined morphologies and enhanced biocompatibility (relative to the physiochemical based nanoparticles) have attracted many scientists to use such resources as nanofactories [16,17]. The bio-based methods for the synthesis of metal nanoparticles are based on the systematic use of plant extracts and microorganisms, like bacteria, yeasts, and fungi [18,19,20,21,22]. In biological synthesis, no capping or stabilizing agents are added as the biomolecules can perform this function themselves [23,24,25]. The properties of these nanoparticles (such as shape, size, etc.), are governed by these biomolecules [13]. These biomolecules also functionalize the nanoparticles, making it more effective relative to nanoparticles synthesized through nonbiological means [17].

The use of plants for biosynthesis has been discussed in detail previously by our own group as well as other scientists across the world [19,20,26,27]. The phytogenic and microgenic biosynthesis of nanoparticles both have their own sets of advantages and disadvantages. Phytogenic synthesis is time economical and relatively simple but usually leads to polydispersed nanoparticles, due to the involvement of diverse phytochemicals like phenols, flavonoids, terpenoids, etc. [8,9]. Moreover, seasonal variations can alter the phytochemical profile of the extracts used for biosynthesis [28]. On the contrary, microbial synthesis is devoid of such disadvantages, however, it requires the maintainance of a sterilized environment and culture conditions, making it relatively complex [29]. Herein, we have discussed the potential of using the microbial world as nanofactories for the biosynthesis of metal nanoparticles or metal based composite materials, both from functional and mechanistic perspectives. Microorganisms hold an exciting amount of potential for the biosynthesis of nanoparticles as the synthesis is eco-friendly and is devoid of the use of hazardous chemicals. Microorganisms are cost effective and do not have high energy requirements. In addition, they can accumulate and detoxify heavy metals through reductase enzymes, which reduce metal salts to their corresponding metal nanoparticle with less polydispersity and a narrow size range [16].

## 3. Bacterial and Cyanobacterial Biosynthesis of MtNPs

Bacterial cells have been readily employed as nanofactories for the synthesis of various metal nanoparticles. Both the extracellular and intracellular approaches have been demonstrated. Extracellular biosynthesis occurs outside the bacterial cell after applying diverse techniques, such as (a) using of the bacterial biomass, (b) using the supernatant of bacterial cultures and (c) using cell free extracts. Extracellular synthesis is preferred over intracellular synthesis because it is devoid of of complex downstream processing [28]. These nanoparticles have been used in a wide array of applications, however mostly for biomedical uses. Recently, silver nanoparticles (AgNPs) that were synthesized using *Bacillus brevis* showed excellent antimicrobial activities against the multidrug resistant strains of *Staphylococcus aureus* and *Salmonella typhi* [30]. *Pseudomonas stutzeri* is another bacterial strain which was found capable of accumulating AgNPs using an intracellular mechanism [31]. The *Bacillus* sp. was also found to synthesize silver nanoparticles in the intracellular periplasmic space [32]. In one study, two different isolated strains of *Pseudomonas aeruginosa* were used for the biosynthesis of gold nanoparticles (AuNPs), generating different sizes of AuNPs [33]. Spherical (10–50 nm) and triangular plate (50–400 nm) AuNPs were produced using *Rhodopseudomonas capsulate* [34]. *Serratia ureilytica* mediated the as-synthesized ZnO nanoflowers that were used on cotton fabrics to provide antimicrobial effects against *S. aureus* and *E. coli* [35]. *Lactobacillus plantarum* was also reported for the biosynthesis of ZnO nanoparticles [36]. *Aeromonas hydrophila* is a gram-negative bacterial strain that is used for the biosynthesis and antimicrobial applications of ZnO nanoparticles [37]. Recently, triangular CuO nanoparticles were produced using *Halomonas elongate*, and their antimicrobial activity was confirmed against *E. coli* and *S. aureus* [38]. In another recent report, super paramagnetic iron oxide nanoparticles (29 nm) were produced using *Bacillus cereus* and they were reported for their anti-cancer effects against the MCF-7 and 3T3 cell lines in a dose-dependent manner [39]. Furthermore, bimetallic Ag-Au nanostructures have also been demonstrated using bacterial strains [40]

Numerous researchers have used cyanobacteria for the biosynthesis of nanoparticles [41]. In one study, eight cyanobacterial strains were screened to investigate their potential for the biosynthesis of AgNPs. The results indicated that seven cyanobacterial strains were capable of producing AgNPs under light, however, only three were capable of producing AgNPs in the dark [42]. The ammonia-sensing potential of the cyanobacterial synthesized silver nanoparticles was also reported [43]. Highly monodispersed AgNPs (5–6.5 nm) were reported through the marine cyanobacterium *Phormidium fragile* [44]. The Nostoc species was also used for the biosynthesis of AgNPs, which indicated significant cytotoxic activity against MCF-7 anti-cancer cell lines, as well as good antimicrobial activities [45]. The intracellular synthesis of AuNPs was performed using *Lyngbya majuscule*, which was isolated from the Arabian Gulf. The authors also revealed the interesting role of the use of AuNPs in combination with *Lyngbya majuscule* as an anti-myocardial infraction agent [46]. 

## 4. Mycosynthesis of Nanoparticles

Myco-nanotechnological approaches have been successfully applied for the biosynthesis of different metal nanoparticles. Likewise, in bacteria/cyanobacteria, the biosynthesis can be intracellular or extracellular. In intracellular synthesis, metal salts are converted into a less toxic form in the mycelia, which can be used by the fungi [47]. Extracellular biosynthesis includes the use of fungal extracts [48]. Fungi are relatively more resourceful then bacteria in the biosynthesis of nanoparticles due to the presence of a number of bioactive metabolites, high accumulation and enhanced production [16,49,50]. Recently, different filamentous fungi were reported to be proficient in the biosynthesis of AuNPs. This study employed different methods for the biosynthesis of AuNPs. The authors proposed that the fungal compounds and fungal media components potentially played a role in stabilizing the nanoparticles [47]. In another research article, three different strains of fungi (namely *Aureobasidium pullulans*, *Fusarium oxysporum* and the *Fusarium* sp.) were used to biosynthesize AuNPs. The authors indicated that the biosynthesis occurred in fungi vacuoles and that reducing sugars were involved to tailor spherical AuNPs. They also established the role of specific fungal proteins in the capping of the AuNPs [51]. A recent study indicated that *Rhizopus stolonifera* extracts mediated the synthesis of monodispersed AgNPs (9.4 nm), while condition optimization resulted in 2.86 nm AgNPs [52]. The extracellular synthesis of AgNPs using *Candida glabrata* indicated good antimicrobial potential [53]. *Aspergillus niger* mediated ZnO nanoparticles indicated excellent antibacterial potential, while they were also able to degrade the Bismarck brown dye by up to 90% [54]. Recently, cobalt oxide nanoparticles were produced using *Aspergillus nidulans* [55]. The nanoparticles were characterized via x-ray diffraction analysis (XRD), transmission electron microscopy (TEM), Fourier transform infrared spectroscopy (FTIR) and energy dispersive x-ray analysis (EDX) techniques, which revealed their spherical shape and average size of 20.29 nm. Interestingly, biogenic cobalt oxide nanoparticles have the potential to be applied in energy storage, lithium-ion batteries and in gas sensing, as well as in medicine.

## 5. Algae as Biosynthesis Factories 

The use of algae is also increasingly popular for the biosynthesis of nanoparticles. *Sargassum muticum* was used for the biosynthesis of ZnO nanoparticles and was reported to decrease angiogenesis along with apoptotic effects in HepG2 cells [56]. AgNPs that were biosynthesized using *Gelidium amansii* indicated excellent antimicrobial properties by forming a diverse biofilm to combat bacterial strains [14]. *Sargassum crassifolium*, a macroalgae and sea weed, has been used in the biosynthesis of AuNPs. Furthermore, the authors observed a blue shift in the UV absorption after increasing the concentration of *S. crassifolium*, which was attributed to a decreased size due to increased nucleation centers in the reductant [57]. *Cystoseira trinodis* was used for the biosynthesis of CuO nanoparticles (7 nm) and was reported to have enhanced antibacterial activities and possess significant potential as an antioxidant, degrading the methylene blue [58]. Aluminum oxide nanoparticles (~20 nm) were produced using *Sargassum ilicifolium* [59]. Different algae strains were reported for the biosynthesis of gold nanoparticles, namely *Turbinaria conoides* [60], *Sargassum tenerrimum* [61], *Acanthophora spicifera* [62], *Laminaria japonica* [63], etc. These have been used for the biosynthesis of AuNPs. Novel core (Au)-shell (Ag) nanoparticle synthesis has also been reported using *Spirulina platensis* [64].

## 6. Mechanisms of MtNP Synthesis by Microorganisms

The synthesis of nanosized substances using microbial cells is an emerging trend in the field of nanotechnology. Microbes including bacteria, fungi, viruses, actinomycetes and yeasts act as potential biofactories for the reduction of silver, gold, gold-silver alloy, cadmium, selenium, magnetite, silica, platinum, titania, palladium and other metals to their subsequent nanoparticles for biological applications [109]. Microbes synthesize these nanoparticles either extracellularly or intracellularly using various bio-reduction processes (Table 2).

## 7. Extracellular Enzymes

Extracellular microbial enzymes are known to play a significant role as reducing agents in the production of MtNPs [110] (Figure 1). Studies suggest that cofactors such as nicotinamide adenine dinucleotide (NADH) and reduced form of Nicotinamide adenine dinucleotide phosphate (NADPH) dependent enzymes both play vital roles as reducing agents via the transfer of the electron from NADH by NADH-reliant enzymes, which act electron carriers [111]. The extracellular synthesis of AuNPs by the bacterium *Rhodopseudomonas capsulata* is mediated via the secretion of NADH and NADH-reliant enzymes. The bioreduction of gold is initiated via electron transfer from NADH by NADH-reliant reductase enzymes present in *R. capsulata*. Consequently, gold ions accept electrons and get reduced (Au^3+^ to Au^0^), leading to the formation of gold nanoparticles [34]. Several other factors, including the concentration of the predecessor, the pH, the temperature and the duration of reaction are limiting factors in controlling the size of MtNPs. Beside these enzymes, several compounds, including naphthoquinones, anthraquinones and hydroquinones are involved in the production of MtNPs [19]. Microbes utilize various mechanisms for the synthesis of NPs, including changes in solubility, biosorption, metal complexation, extracellular precipitation, toxicity via oxidation-reduction, the absence of specific transporters and efflux pumps [19,112].

Several fungi produce extracellular enzymes like acetyl xylan esterase, cellobiohydrolase D, glucosidase and β-glucosidase, which are known to play a significant role in the biosynthesis of MtNPs [15]. One mechanism involved in the extracellular synthesis of AgNPs is the use of nitrate reductase which is secreted by fungi, which helps in the bioreduction and synthesis of MtNPs. Several studies reported the involvement of nitrate reductase in the extracellular synthesis of MtNPs [113,114]. Studies involving the use of commercially available nitrate reductase disks revealed that these NADH-reliant reductase enzymes were involved in the reduction of Ag^+^ ions to Ag^(0)^ and the subsequent formation of silver nanoparticles [115,116]. *Fusarium oxysporum* was used as source of reducing agents for the synthesis of gold and silver NPs. Results showed that extracellular reductases produced by the fungi caused the reduction of Au^3+^ and Ag^1+^ to Au–Ag alloy NPs. Moreover, nitrate-reliant reductases and shuttle quinone obtained from various species of this fungi were utilized in the extracellular synthesis of NPs [117]. However, some species like *Fusarium moniliforme* failed to generate AgNPs, even upon the release of the reductase, indicating the Ag^1+^ reduction via the involvement of conjugated oxidation-reduction reactions of electron carriers involving NADP-reliant nitrate reductase [116]. Furthermore, nitrate reductase from *F. oxysporum* was used in an in vitro study in oxygen free conditions in the presence of a cofactor (NADPH), a stabilizer protein (phytochelatin) and an electron carrier (4-hydroxyquinoline) in order to synthesize AgNPs. This fungi exhibited good extracellular production of AgNPs and can be considered an excellent candidate for the extracellular synthesis of other MtNPs [113,118]. Yet in other studies, *F. oxysporum* was used for the extracellular synthesis of semiconductor CdS nanoparticles, where extremely luminescent CdSe nanoparticles were synthesized using the reductase enzyme of the fungi [114,119]. Enzymes from other fungal stains, including *Fusarium semitectum* and *Fusarium solani*, were used for the extracellular production of AgNPs. The results of the study revealed that specific proteins might be responsible for the reduction of Ag^+^, thus forming AgNPs [120,121]. *Cladosporium cladosporioides* and *Coriolus versicolor* were effectively used for the extracellular synthesis of AgNPs that involved fungal proteins, organic acids and polysaccharides which effect the growth and shape of the nanocrystals [122]. Subsequent to incubation of *Aspergillus niger* in a AgNO_3_ solution, the extracellular production of AgNPs was stabilized by fungal proteins [123]. Likewise, *Aspergillus fumigatus* extracellularly produced AgNPs in exceptionally less time (10 min), as compared to other physical and chemical techniques [124]. Thus, *A. fumigatus* is an ideal candidate for industrial scale production of a variety of NPs. *Penicillium fellutanum* was also observed to reduce Ag^1+^ ions in a very short amount of time (10 min). Further studies revealed that a protein of nitrate reductase was responsible for the reduction of Ag^1+^ ions [125]. *Penicillium brevicompactum* was reported to cause the reduction of Ag^1+^ ions via the liberation of NADH-reliant enzyme nitrate reductases [126].

Nanotechnology was also applied to a new group of the plant kingdom known as algae. Among the algae, *Sargassum wightii greville* was reported to rapidly reduce Au^3+^ ions to form AuNPs 8–12 nm in size [127]. Another filamentous algae, *Chlorella vulgaris*, was used in the biosynthesis of gold nanoparticles, resulting in the formation of Au and Au^+1^S nanoparticles [128]. 

## 8. Intracellular Enzymes

In the intracellular mechanism of metal bioreduction, bacterial and fungal cells along with sugars molecules play a significant role. Mainly, the interactions of intracellular enzymes and positively charged groups are utilized in the gripping of metallic ions from the medium and the subsequent reduction inside the cell [150,151]. When observed microscopically, MtNPs accumulated in the periplasmic space, the cytoplasmic membrane and the cell wall. This was due to the diffusion of metal ions across the membranes and enzymatic reduction resulting in MtNPs.

Among the actinomycetes, alkalo-tolerant (*Rhodococcus* sp.) and alkalo-thermophilic (*Thermomonospora* sp.) actinomycetes were used for the intracellular synthesis of AuNPs [148,152]. The intracellular production of AuNPs with uniform dimensions was carried out by reacting the *Rhodococcus* species with an aqueous solution of AuCl^4−^ ions. Reductions of Au^3+^ were effectively mediated by enzymes at the surface of the mycelia and the cytoplasmic membrane. The exposure of a *Verticillium* biomass to a Ag^+^ ionic solution resulted in intracellular reduction and the subsequent formation of AgNPs. Visualization via electron microscopy revealed that the AgNPs were formed under the surface of cell wall as a result of enzymatic bioreduction, which is non-toxic to the fungi [153]. The same procedure was adopted for the synthesis of AuNPs using the fungus *Verticillium* as the source of reducing enzymes. AuNPs were entrapped in the cytoplasmic membrane and the cell wall of the fungi, indicating that Au^3+^ was bio-reduced by reductase enzymes that were present there [154]. Southam and Beveridge reported that AuNPs were formed and precipitated inside bacterial cells after the incubation of the bacterial cells in a Au^3+^ ionic solution [155]. *Pseudomonas stutzeri* (AG259) when exposed to concentrated AgNO_3_ solution has reduced Ag^1+^ ions, with the subsequent formation of AgNPs in the bacterial periplasmic space [31]. A filamentous cyanobacterium (*Plectonema boryanum*) treated with AuCl^4−^ and Au(S_2_O_3_)_2_^3-^ solutions resulted in the formation of AuNPs at the membrane level and gold sulfide residing intracellularly [128]. In another study, *Phanerochaete chrysosporium* incubation in an ionic Au^3+^ solution resulted in the formation of AuNPs 10–100 nm in particle size. The Laccase enzyme was used as an extracellular reducing agent, whereas ligninase was found to be responsible for the intracellular reduction of Au^3+^ ions [156]. Other factors, including the incubation age of the fungi, the concentration of the AuCl^4−^ solution and the incubation temperature demonstrated significant effects on the shape of AuNPs. The mesophilic bacterium *Shewanella algae* proved to be an efficient bioreducer of AuCl^4−^ ions to elemental gold. Au nanoparticles were found in the bacterial periplasmic space, mediated via intracellular enzymes [157]. *Brevibacterium casei* treated with aqueous solutions of Au^3+^ and Ag^+^ ions were reduced to intracellular enzymes, mediated by spherically shaped AuNPs and AgNPs respectively [147].

## 9. Challenges and Limitation of MtNP Synthesis by Microorganisms: A Possible Solution

Nanotechnology is an interdisciplinary field which involves close coordination between engineers, chemists and biologists to design and develop potential candidates for therapeutic applications. Research on nanoparticle biosynthesis using microbes has gained significant interest, owing to its simplicity, low cost of production and eco-friendliness. MtNPs have emerged as excellent agents for biomedical applications due to their unique physicochemical, optical, electronic and biological properties [158]. Expanding interest on the application of MtNPs in biology and healthcare has raised a need for the development of cheap, easy and eco-friendly methods for MtNP preparation, such as the bioinspired approaches [159,160]. The biosynthesis of MtNPs using microbes is well investigated by researchers globally, where the reduction of metal ions to MtNPs occurs as part of the heavy metal-resistance mechanism exhibited by microorganisms [161]. The primary challenge involved in microbial inspired synthesis is the selection of the best candidates, based on the intrinsic properties of microbes, such as growth rate, replication, biochemical activity etc. Controlling the shape, size and monodispersity of the biosynthesized MtNPs is another important criterion for achieving the desired therapeutic effects, which might be modulated by varying the concentration of biomolecules, the reduction time, the temperature and other factors. Another important challenge involved in biosynthesis is the identification of the key components among the large pool of biomolecules obtained from microbial resources responsible for the reduction and stabilization of MtNPs from precursor salts. Maintaining the optimal conditions needed for the microbial growth and enzymatic activity is critical as these act as biocatalysts, playing a major role in the preparation of MtNPs. Optimizing the conditions, such as providing essential nutrients for growth, the size of inoculum, the temperature, the pH and the amount of light greatly enhance the efficiency of MtNP biosynthesis using microbes [162]. Yield of the final product and rate of biosynthesis of MtNPs using microorganisms are another set of important parameters to be considered to translate the measure of production to a large scale industrial level. In general, the large scale production of biosynthesized nanoparticles using microbes has several challenges in scaling up, due to medium to low yield compared to chemically synthesized nanoparticles. In general, the yield of the biosynthesis methods are 1/3rd of the yield obtained from the chemical synthesis methods. Also, we have to keep in mind that the laboratory scale bottom-up or top-down approaches for the synthesis of MtNPs varies significantly from industrial manufacturing. However, due to large availability, easy access and the high growth rate of microbes, the overall cost of the synthesis will be reduced at a large extent. Moreover, the cost of the biosynthesis gets further reduced as there is no requirement of organic solvents, chemical stabilizing agents, thermal heat or any fancy technique during these synthesis methods. Hence, it can be safely mentioned that, if the right techniques are properly utilized, the overall production cost of the NPs can be reduced by almost 1/10th compared to the chemical synthesis methods in the long run. Nevertheless, we strongly feel that during the initial stages of the development of NPs at the lab scale, researchers must consider and deploy appropriate methods useful for large-scale industrial manufacturing [163]. An advantage of this method is that the MtNPs produced using the biosynthesis route are more stable without any agglomeration, even at room temperatures for long periods of time when compared with MtNPs produced using chemical synthetic methods [164,165]. This stability might be due to the in situ capping of the microbial proteins or biomolecules over the MtNP surface during the biosynthesis process. Understanding the source and mechanism of biosynthesis is essential, as sometimes the utility of toxic microorganisms as a bioresource might impart pathogenicity to the prepared MtNPs. Above all, the protocols designed at the laboratory scale might yield well-characterized highly stable MtNPs. Scaling up this process to the industrial level is mandatory to fulfill the unmet medical challenges, which mainly depend upon optimizing the growth rate of the microbes, inoculation of the seed into the biomass, harvesting cells and effective MtNP preparation through reduction, as well as recovery of the final product with good yield [12,166,167]. Taken together, there are several important challenges and technical aspects that must be addressed to successfully translate this bioinspired method of MtNP synthesis for large scale industrial production.

## 10. Conclusions and Future Prospects

For the time being, it can be safely concluded that there is a bright future for the microbial synthesis of metal nanoparticles and their wide variety of applications in biomedical nanotechnology when compared with chemically assisted NPs, due to their low cost and eco-friendliness. The huge pool of bioresources (microbes and microbial enzymes), if appropriately exploited, could help biosynthesized NPs to turn out to be a potential game changer in the imminent future. However, several challenges must be dealt with before industrial scale production and wide use, which will take a little more time (5–10 years). Nevertheless, there is immense potential for microbial assisted metal nanoparticles as they are low in toxicity and cost and high in degradability and are useful in numerous therapeutic applications. It is also encouraging that several groups are concentrating on decoding the detailed mechanistic aspects of microbial biosynthesis, which will ultimately lead to better understanding and wiser applicability. Even though the mechanisms behind these nanoparticles’ uptake, diffusion, long term toxicity and excretion remain uncertain, the potential biological applications make them a potential candidate to extensively substitute chemically synthesized NPs in the coming years.

## Figures and Tables

**Figure 1 ijms-19-04100-f001:**
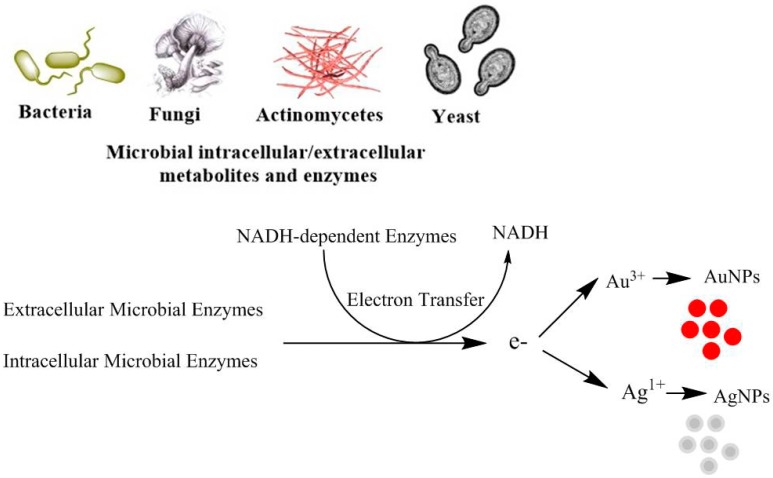
Role of NADH and NADH-dependent microbial enzymes in the synthesis of metal nanoparticles (MtNPs).

**Table 1 ijms-19-04100-t001:** Studies from 2016 to date, indicating different microorganisms for nanoparticle biosynthesis.

Serial Number	Microorganisms	Nanoparticle	Size/Shape	Application	Reference
**Bacteria**					
1	*Actinobacter*	Ag	13.2 nm/Spherical	Antibacterial	[65]
2	*Acinetobacter*	Au	19 nm/Spherical-triangular-polyhedral	-	[66]
3	*Klebsiella pneumonia*	Au	10–15 nm/Spherical	Antibacterial	[67]
4	*Sinomonas mesophila*	Ag	4–50 nm/Spherical	Antibacterial	[68]
5	*Pseudomonas fluorescens*	Au	5–50 nm/Spherical	Antibacterial	[69]
6	*Bacillus endophyticus*	Ag	5.1 nm/Spherical	Antimicrobial	[70]
7	*Bacillus brevis*	Ag	41–68 nm/Spherical	Antibacterial	[30]
8	*Streptomyces* *griseoplanus*	Ag	19.5–20.9 nm/Spherical	Antifungal	[71]
9	*Nocardiopsis flavascens*	Ag	5 and 50/Spherical	Cytotoxicity	[72]
10	*Caldicellulosiruptor changbaiensis*	Au	<20 nm/Spherical	Antibacterial, Antibiofilm	[73]
11	*Shewanella loihica*	Cu	10–16 nm/Spherical	Antibacterial	[74]
12	*Shewanella loihica*	Pt	1–10 nm/Spherical	Dye degradation	[75]
13	*Shewanella loihica*	Pd	1–12 nm/Spherical	Dye degradation	[75]
14	*Shewanella loihica*	Au	2–15 nm/Spherical	Dye degradation	[75]
15	*Micrococcus yunnanensis*	Au	53.8 nm/Spherical	Antibacterial, Anticancer	[76]
16	*Mycobacterium* sp.	Au	5–55 nm/Spherical	Anticancer	[77]
17	*Halomonas salina*	Au	30–100 nm/Spherical	-	[78]
**Fungi**					
18	*Aspergillus niger*	ZnO	53–69 nm/Spherical	Antibacterial Dye degradation	[54]
19	*Trametes trogii*	Ag	5–65 nm/Spherical- Ellipsoidal	-	[79]
20	*Trichoderma longibrachiatum*	Ag	10 nm/Spherical	Antifungal against phyto-pathogenic fungi	[80]
21	*Trichoderma harzianum*	Au	32–44 nm/Spherical	Antibacterial, Dye degradation	[81]
22	*Fusarium oxysporum*	Ag	21.3–37 nm/Spherical	Antimicrobial	[82]
23	*Pleurotus ostreatus*	Au	10–30 nm/Spherical	Antimicrobial, Anticancer	[83]
24	*Aspergillus terreus*	Ag	16–57 nm/Spherical	Antibacterial	[84]
25	*Ganoderma sessiliforme*	Ag	~45 nm/Spherical	Antibacterial, Antioxidant, Anticancer	[85]
26	*Phenerochaete chrysosporium*	Ag	34–90 nm/Spherical-Oval	Antibacterial	[86]
27	*Penicillium polonicum*	Ag	10–15 nm/Spherical	Antibacterial	[87]
28	*Candida glabrata*	Ag	2–15 nm/Spherical	Antibacterial	[53]
29	*Macrophomina phaseolina*	Ag/AgCl	5–30 nm/Spherical	Antibacterial	[88]
30	*Aspergillus nidulans*	CoO	20.29 nm/Spinel	-	[55]
31	*Rhodotorula glutinis*	Ag	15.45 nm/Spherical	Antifungal, Dye degradation, Cytotoxicity	[89]
32	*Rhodotorula* *mucilaginosa*	Ag	13.70 nm/Spherical	Antifungal, Dye degradation, Cytotoxicity	[89]
33	*Cladosporium* sp.	Ag	24 nm/Spherical	Antioxidant, Antidiabetic, Anti-Alzheimer	[90]
34	*Cladosporium cladosporioides*	Au	60 nm/Round	Antioxidant, Antibacterial	[91]
35	*Nemania* sp.	Ag	33.52 nm/Spherical	Antibacterial	[92]
36	*Penicillium chrysogenum*	Pt	5–40 nm/Spherical	Cytotoxicity	[93]
37	*Aspergillus* sp.	Au	2.5–6.7 nm/Spherical	Nitrophenol reduction	[94]
38	*Rhizopus stolonifer*	Ag	2.86 nm/Spherical	-	[52]
**Algae/Cyanobacteria**					
39	*Sargassum wightii*	ZrO_2_	18 nm/Spherical	Antibacterial	[95]
40	*Neochloris oleoabundans*	Ag	40 nm/Spherical	Antibacterial	[96]
41	*Cystoseira baccata*	Au	8.4 nm/Spherical	Anticancer	[97]
42	Stephanopyxis turris	Au	10–30 nm/Spherical	-	[98]
43	*Galaxaura elongate*	Au	3.85–77 nm/Spherical-rods-triangular	Antibacterial	[99]
44	*Chlorella vulgaris*	Pd	5–20 nm nm/Spherical	-	[100]
45	*Enteromorpha compressa*	Ag	4–24 nm/Spherical	Antimicrobial, Anticancer	[101]
46	*Nostoc linckia*	Ag	5–60 nm/Spherical	Antibacterial	[102]
47	*Nostoc sp*	Ag	51–100 nm/Spherical	Spherical	[45]
48	*Leptolyngbya*	Ag	5–50 nm/Spherical	Antibacterial, Anticancer	[103]
49	*Spyridia fusiformis*	Ag	5–50 nm/Spherical	Antibacterial	[104]
50	*Chlorella pyrenoidosa*	CdSe QD	4–5 nm	Imatinib sensing	[105]
52	*Sargassum ilicifolium*	Al_2_O_3_	20 nm/Spherical	-	[59]
53	*Padina pavonia*	Ag	49.58–86.37 nm/spherical-triangular-rectangle-polyhedral-hexagonal	-	[106]
53	*Spirulina platensis*	Pd	10–20 nm/Spherical	Adsorbent	[107]
54	*Chlorella pyrenoidosa*	TiO_2_	50 nm/Spherical	Dye degradation	[108]

**Table 2 ijms-19-04100-t002:** List of extracellular and intracellular bioreducing microbial enzymes and resulting nanoparticles.

**Source of Extracellular Enzymes (Bacterial Species)**	**Nature of Organism**	**Metals Used**	**Shape**	**Size (nm)**	**Temperature (°C)**	**Reference**
*Desulfovibrio desulfuricans*	G−ive Bacteria	Pd	Round	50	25	[129]
*Pyrobaculum islandicum*	G−ive Rods	U, Tc, Cr, Co, Mn	Round	NA	100	[130]
*Escherichia coli*	G−ive Bacteria	CdTe	Round	2–3.2	37	[131]
*Escherichia coli*	G−ive Bacteria	Au	Hexagonal, Triangle	20–30	37	[132]
*Bacillus licheniformis*	G+ive mesophilic bacteria	Ag	NA	50	37	[133]
*Shewanella* species	Marine Bacteria	Se	Round	181	30	[134]
*Ureibacillus thermosphaericus*	G+ive Bacteria	Au	NA	50–70	60–80	[135]
*Corynebacterium glutamicum*	G+ive Bacteria	Ag	Irregular	5–50	30	[136]
*Rhodopseudomonas capsulate*	Phototrophic Bacteria	Au	Round	10–20	30	[34]
*Pseudomonas aeruginosa*	G−ive Bacteria	Au	NA	15–30	37	[33]
*Shewanella Oneidensis*	Facultative Bacteria	Au	Round	12	30	[137]
**Fungi and Algae Species**						
*Plectonema boryanum* UTEX 485	Filamentous Fungi	Au	Octahedral	10 nm–6 µm	25	[138]
*Phaenerochaete chrysosporium*	Fungi	Ag	Pyramidal	50–200	37	[139]
*Aspergillus flavus*	Fungi	Ag	Round	8.92	25	[140]
*Yeast*	Fungi	Au, Ag	Polygonal	9–25	30	[131]
*Fusarium oxysporum*	Ascomycete fungus	Alloy of Au–Ag	Round	8–14	25	[117]
*Sargassum wightii*	Macro-algae	Au	Planar	8–12	NA	[127]
*Neurospora crassa*	Bread mold	Au–Ag, Au	Round	20–50	28	[50]
*Verticillium* sp.	Fungi	Ag	Round	25–32	25	[141]
*Aspergillus fumigatus*	Fungi	Ag	Round	5–25	25	[124]
*Trichoderma viride*	Fungi	Ag	NA	2–4	10–40	[142]
*Yarrowia lipolytica*	Fungi	Au	Triangles	15	30	[143]
**Source of Intracellular Enzyme (Bacterial Species)**	**Nature of Microb.**	**Metals used**	**Shape**	**Size** **(nm)**	**Temperature (°C)**	**Reference**
*Shewanella algae*	G−ive marine bacteria	Pt	NA	5	25	[144]
*Enterobacter* species	Anaerobic G−ive Bacilli	Hg	Round	2–5	30	[145]
*Bacillus cereus*	G+ive Bacteria	Ag	Round	4–5	37	[146]
*Brevibacterium casei*	Actinomycetales Bacteria	Ag, Au	Roud	10–50	37	[147]
*Rhodococcus* sp.	Actinobacteria	Au	Round	8-12	NA	[148]
**Fungi and Algae Species**						
*Plectonema boryanum*	Algae	Au	Cubic	<10–25	25–100	[128]
*Neurospora crassa*	Bread mold	Au–Ag, Au	Round	32	28	[50]
*Verticillum luteoalbum*	Ascomycota Fungi	Au	NA	NA	37	[149]
*Candida utilis*	Fungus	Au	NA	NA	25	[149]

NA: Not available; G+ive: Gram-positive; G−ive: Gram-negative.

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
