# Peer review of "Biosynthesis of Metal Nanoparticles via Microbial Enzymes: A Mechanistic Approach"

_ijms, 2018, doi:10.3390/ijms19124100_

Round 1
Reviewer 1 Report
This is a very complex and interesting review report which show the role of biosynthesis in the production of nanoparticles. Authors use the abbreviation MNP - this is usually used for magnetic nanoparticles, not for metallic nanoparticles.
Author Response
We are thankful to reviewer's positive comments on our review article. According to reviewer suggestion we have now replaced the metallic nanoparticle abbreviation by MtNPs instead of MNPs.
Reviewer 2 Report
The manuscript is well-written and would be of interest to the audience of journal. Therefore, it is recommended that this contribution be accepted as well as.
Author Response
We are grateful to reviewer for proving positive feedback on our review article and recommending to accept for publication.
Reviewer 3 Report
A well written and comprehensive review on the use of microbes for inorganic nanoparticle production. This is very timely for the field and I believe will be of interest to the readers. In my opinion however, it was a little one sided and I would like more information included about the current drawbacks of such production. There was only a fleeting mention of yield being an important factor to consider. I would like to see some more information about current yield ability, how does this rival the chemical synthesis, what are the realistic cost implications. I think this manuscript would benefit greatly from inclusion of this information.
Author Response
We are thankful to reviewer for his/her constructive suggestions. Accordingly, we have now elaborated the comparison between chemical synthesis and microbe mediated synthesis of NPs with respect to yield, cost and other important parameters.